# Effectiveness of Double Balloon Enteroscopy in the Diagnosis and Treatment of Small Bowel Varices

**DOI:** 10.3390/diagnostics15030336

**Published:** 2025-01-31

**Authors:** Suleyman Dolu, Mehmet Emin Arayici, Soner Onem, Ilker Buyuktorun, Huseyin Dongelli, Goksel Bengi, Mesut Akarsu

**Affiliations:** 1Department of Gastroenterology, Faculty of Medicine, Dokuz Eylül University, Izmir 35340, Turkey; ilker_bt@hotmail.com (I.B.); drgokselbengi@hotmail.com (G.B.); 2Department of Biostatistics and Medical Informatics, Faculty of Medicine, Dokuz Eylül University, Izmir 35340, Turkey; mehmet.e.arayici@gmail.com; 3Department of Gastroenterology, Samsun Training and Research Hospital, Samsun 55090, Turkey; soneronem@hotmail.com; 4Department of Internal Medicine, Faculty of Medicine, Dokuz Eylül University, Izmir 35340, Turkey; juniordongel@gmail.com; 5Private Gastroenterology Clinic, Izmir 35535, Turkey; mesutakarsu71@gmail.com

**Keywords:** double balloon enteroscopy, small bowel, ectopic varices, endotherapy, diagnosis

## Abstract

**Background/Aims:** Double balloon enteroscopy (DBE) is an innovative method for the diagnosis and management of small bowel (SB) diseases. SB varices are rare disorders, and their diagnosis and treatment can be challenging for clinicians. This study evaluates the use of double balloon enteroscopy (DBE) in diagnosing and treating small bowel varices. **Materials and Methods:** SB varices were detected in 28 out of 900 double balloon enteroscopy procedures over an 18-year period. Eleven cases of SB varices of various etiologies, diagnosed via DBE, are described. The characteristics of SB varices and endoscopic procedural details were evaluated. **Results:** A retrospective investigation of 750 patients identified eleven patients (eight males and three females; median age 59 years, range 40–80 years) with small bowel varices. The most common site of SB varices was the jejunum. At least one abdominopelvic surgical procedure had been previously performed on five patients. Endotherapy by DBE was administered to nine patients (seven emergent and two prophylactic). Post-endotherapy, three patients experienced bleeding that required re-endotherapy. Endoscopic therapy for small bowel varices included injection sclerotherapy in eight cases (six with cyanoacrylate and two with polidocanol) and injection sclerotherapy plus hemoclipping in one case. **Conclusions:** SB varices can present a diagnostic challenge for clinicians. DBE is a valuable tool for both the diagnosis and management of small bowel varices.

## 1. Introduction

Ectopic varices are dilated portosystemic collateral veins located outside the gastroesophageal junction and represent a less common but clinically significant manifestation of portal hypertension [1,2]. These varices can occur in various anatomical locations, but two-thirds are found in the small bowel (SB), making this site the most prevalent site for ectopic varix. While cirrhosis-related portal hypertension and segmental or diffuse mesenteric venous congestion resulting from surgical sequelae are the most common causes of SB varices, thrombosis and/or malignancy are also well-known causes. Clinicians often face challenges in diagnosing bleeding ectopic varices located distal to the ligament of Treitz because of their deep and atypical location within the gastrointestinal tract [3,4,5]. Advanced diagnostic modalities such as capsule endoscopy, device-assisted enteroscopy (DAE), computed tomography (CT), CT angiography, and, in some cases, surgical exploration play crucial roles in identifying and localizing these elusive lesions [1,6].

The small bowel, measuring approximately 5–7 m in length, accounts for 75% of the total length of the gastrointestinal tract, highlighting its extensive and complex structure [7,8]. Double balloon enteroscopy (DBE) is an advanced endoscopic technique designed to provide detailed visualization and access to the entire small bowel, making it an indispensable tool for various clinical indications, such as the detection and management of polyps and tumors [7,8]. Beyond its role in diagnosing neoplasms, DBE is also instrumental in identifying and treating vascular diseases, including those with bleeding complications [9,10]. Despite its broad clinical utility, the literature on the use of DBE specifically for diagnosing and managing small bowel varices remains limited, with most publications being case reports rather than large-scale studies, underscoring the rarity and diagnostic challenges associated with these lesions [11].

Although rare, bleeding ectopic varices constitute less than 5% of all variceal hemorrhages, presenting a significant diagnostic and therapeutic challenge for clinicians [1]. When bleeding occurs in SB varices, effective management often requires a multidisciplinary approach that combines endoscopic, radiological, and surgical techniques tailored to the patient’s condition. Endoscopic treatments, including injection therapy with sclerosant agents, cyanoacrylate, and endoscopic band ligation, are employed as minimally invasive options to control bleeding. However, the lack of a clear consensus on the superiority of one endoscopic technique over another reflects the variability in clinical practice and the need for individualized treatment strategies based on variceal location, size, and patient comorbidities [11,12,13,14]. In addition to endoscopic interventions, interventional radiological procedures, such as transjugular intrahepatic portosystemic shunt placement, balloon-occluded retrograde transvenous obliteration, venous dilatation, stent placement for occluded veins, and embolization therapy, offer alternative and often complementary methods for managing bleeding SB varices [12].

The diagnosis and management of SB varices pose significant challenges for clinicians, largely owing to the scarcity of comprehensive data in the existing literature, which is predominantly composed of isolated case reports and small case series. This limitation highlights the need for further investigation and an evidence-based approach. Against this backdrop, the present study aimed to assess the diagnostic and therapeutic effectiveness of DBE in identifying and managing SB varices. By focusing on the capabilities of DBE, this study aimed to provide valuable insights into its role as a diagnostic and interventional tool, potentially improving outcomes for patients with this rare and complex condition.

## 2. Materials and Methods

This retrospective and descriptive study was conducted at Dokuz Eylül University Hospital, analyzing 900 DBE procedures performed on 750 patients over an 18-year period, from January 2006 to April 2024. Among these, SB varices were identified endoscopically in 28 DBE procedures involving 11 patients. Comprehensive demographic, clinical, endoscopic, and radiological data were collected and analyzed to provide insights into this rare condition. The characteristics of SB varices, including their location, number, and symptoms leading to diagnosis, were systematically documented, along with detailed procedural data, such as diagnostic findings, therapeutic interventions, procedural approaches, and complications. SB varices were categorized by location into three groups: duodenum, jejunum, and ileum. Patients were further stratified based on the underlying etiology of SB varices and classified into four categories: postsurgical adhesion, portomesenteric venous thrombosis, cirrhosis, and unknown etiology. Importantly, patients with portomesenteric venous thrombosis were excluded from any other etiological group to avoid overlap. All patients underwent thorough radiological and laboratory evaluations to assess portal hypertension, with and without accompanying portomesenteric venous thrombosis. Intra-abdominal adhesions from previous abdominal surgeries can predispose patients to the development of collaterals in unusual locations, leading to ectopic varices [1].

In this study, double balloon enteroscopy played a central role as a diagnostic and therapeutic tool. However, radiological modality, such as CT or CT angiography, was often utilized as a complementary tool to identify varices and their anatomical locations. In addition, capsule endoscopy could not be used for a diagnostic tool, because it was not available in our hospital. The study population comprised patients who underwent DBE for a range of indications, including gastrointestinal bleeding and abnormal findings on radiological imaging. In this study, we specifically included patients with SB varices diagnosed using DBE.

Our hospital serves as a tertiary referral center for DBE, attracting a diverse patient population with various gastrointestinal conditions. The study population comprised patients who underwent DBE for a range of indications, including gastrointestinal bleeding and abnormal findings on radiological imaging. In this study, we specifically included patients with SB varices diagnosed using DBE. Patients who did not undergo DBE or were under 18 years of age were excluded to ensure a focused and consistent analysis of this unique cohort. This selection criterion allowed for a comprehensive evaluation of the diagnostic and therapeutic roles of DBE in managing SB varices.

In all cases, SB varices were diagnosed using DBE, a highly specialized endoscopic technique. The DBE system used for the procedures consisted of an overtube and enteroscope, pumping unit, and main computer. All procedures were performed using the Fujinon EN-450T5 enteroscope (Fujinon Corp., Saitama, Japan), a device specifically designed for deep and comprehensive visualization of the small bowel. For the endoscopic treatment of varices, sclerosing agents such as cyanoacrylate and polidocanol were administered under direct visualization. The exact amount of the agent used varied based on the size, location, and number of varices. Cyanoacrylate was typically injected in volumes ranging from 0.5 to 1.0 mL per varix, diluted with lipiodol to ensure adequate radiopacity and effective distribution within the varix. Polidocanol, on the other hand, was administered as a 1% solution in volumes of 1–2 mL per varix, depending on its dimensions and extent of bleeding. After the injection, the treated area was inspected for hemostasis, and additional measures, such as hemoclipping, were applied if needed. Prior to the DBE examination, written informed consent was obtained from all patients to ensure ethical compliance and patient understanding of the procedure. An experienced endoscopist, with 18 years of expertise and over 1500 DBE procedures, conducted all examinations, guaranteeing a high level of technical proficiency. Patient safety was prioritized through a comprehensive pre-procedure assessment by an anesthesiologist who determined each patient’s suitability for sedation. During the DBE procedures, the patients were accompanied by an anesthesiologist and received either deep or conscious sedation, depending on their medical condition and procedural requirements. Continuous monitoring was performed throughout the procedure to ensure optimal safety and comfort.

### 2.1. Ethical Statements

This study was conducted in accordance with the ethical principles outlined in the Declaration of Helsinki, ensuring the protection of the participants’ rights, safety, and well-being. Ethical approval for the study was obtained from the Dokuz Eylül University Faculty of Medicine Non-Invasive Clinical Research Ethics Committee (approval number: 2024/16-36; date of approval: 8 May 2024).

### 2.2. Statistical Analysis

Descriptive statistics were used to summarize the data. Continuous variables, such as patient age, are presented as medians with ranges because of the small sample size and non-normal distribution. Categorical variables, including etiology, location, and treatment modalities of small bowel varices, were expressed as frequencies and percentages. No inferential statistical tests were applied because of the limited sample size and the retrospective nature of the study. The results are reported descriptively to provide insights into the clinical and procedural characteristics of patients with small bowel varices and their outcomes after double balloon enteroscopy. Statistical analyses were utilized using the IBM SPSS Statistics for Mac, Version 29.0 (IBM Corp., Armonk, NY, USA) package program.

## 3. Results

A total of 900 DBEs were performed from January 2006 to April 2024. The flowchart of the search strategy is shown in Figure 1. Eleven patients with SB varices were identified and most commonly located in the jejunum. Of these patients, eight were men and three were women, with a median age of 59 years (range 40–80 years). The underlying etiologies of the SB varices varied. Adhesions resulting from surgery were identified in three patients (27.3%), including cases with gastrojejunostomy (9.1%), hepaticojejunostomy (9.1%), and subtotal colectomy (9.1%). The reasons for surgery in these cases were gastric outlet obstruction, liver alveolar hydatid cysts, and diverticulosis. Portomesenteric venous thrombosis was the most common etiology, affecting six patients (54.5%). Within this group, two patients (18.2%) had a history of liver transplantation, while the others had conditions such as pancreatic cancer (9.1%), pancreatitis (9.1%), and thrombophilia (9.1%). Among the portomesenteric venous thrombosis cases, some patients exhibited isolated portal vein thrombosis, while others had concurrent mesenteric vein thrombosis.

Specifically, two patients with a history of liver transplantation and alcohol use presented with isolated portal vein thrombosis, whereas two patients with pancreatic cancer and thrombophilia had both portal and mesenteric vein thrombosis. Additionally, one patient (9.1%) had cirrhosis-related portal hypertension, while the etiology remained unknown for one patient (9.1%). In total, three patients had liver cirrhosis, with two having undergone liver transplantation. The etiologies of cirrhosis in the transplanted patients were hepatitis B virus infection and alcohol use, while the cirrhosis in the non-transplanted patient was due to hepatitis B virus infection. The etiologies of the SB varices are summarized in Table 1 and Table 2. Two patients with SB varices had a history of esophageal varices and had received prior treatment.

Twenty-eight DBEs (twenty-six oral and two anal) were performed by experienced interventional endoscopists during the study period. No acute adverse events occurred during these procedures. The indication for enteroscopy was bleeding in ten patients and an abnormal radiological sign in one patient. Endoscopic therapy was not pursued in one patient who underwent DBE due to abnormal radiological findings. Additionally, prophylactic endotherapy was considered for ileal varices in one patient; however, these varices were not observed on a follow-up evaluation. The remaining nine patients received endotherapy. Endoscopic treatment for ectopic varices included injection sclerotherapy alone in eight cases (six treated with cyanoacrylate and two with polidocanol). One patient received both endoscopic injection sclerotherapy with polidocanol and hemoclipping. Band ligation was not utilized for ectopic varices, nor were interventional radiological or surgical treatments administered. Endoscopic images depicting treatments applied to small bowel varices are presented in Figure 2.

Among the patients who received endotherapy, the procedure was emergent in seven cases and prophylactic in two. Post-endotherapy, three patients experienced massive bleeding within one month; two of these patients were managed with a single endotherapy session, while the third required multiple endotherapies and ultimately succumbed to hypovolemic shock. The ten surviving patients had a median follow-up of two years, during which no recurrent gastrointestinal bleeding was observed in eight patients (repeat control DBE procedures were conducted in four patients, with three ectopic varices disappearing post-treatment). Patients who had stigmata but no active bleeding underwent endoscopic treatment in another session because histoacryl was not available in our hospital at the time of diagnosis. Therefore, some patients underwent repeated enteroscopy after histoacryl was obtained. However, the patients with active bleeding underwent endoscopic treatment with polidocanol. In addition, when DBE was performed in these patients, capsule endoscopy could not be used for a diagnostic tool, because it was not available in our hospital. Patients with suspected small bowel bleeding underwent DBE again. Additionally, a repeat DBE was performed on one patient with varices located at two different sites. One patient underwent DBE in an emergency operating room, which was the only death due to bleeding. In our study, only one patient died of severe bleeding due to small bowel varices despite undergoing endoscopic intervention. This patient had no alternative therapies such as radiological or surgical interventions. This case involved significant hemodynamic instability, which precluded a safe transfer to interventional radiology. Furthermore, the patient’s poor overall prognosis led to a focus on endoscopic treatment rather than high-risk surgery. Table 3 summarizes the characteristics of the patients’ DBE treatments.

## 4. Discussion

The diagnosis and management of SB varices remain challenging because of their deep and atypical localization within the gastrointestinal tract. DBE has emerged as a valuable technique for diagnosing and managing rare lesions. In this study, we evaluated 11 patients with SB varices diagnosed using DBE and highlighted the diagnostic accuracy and therapeutic potential of this approach. Among the interventions performed, one patient was successfully treated with a hemoclip, demonstrating the feasibility and effectiveness of endoscopic intervention in managing SB varices. These findings underscore the role of DBE as a crucial tool for addressing the challenges associated with SB varices.

Varices are dilated portosystemic venous collateral veins most commonly found in the gastroesophageal region. Varices occurring outside the esophageal and gastric regions are collectively referred to as ectopic varices [1,15]. The literature shows variability in the reported prevalence of ectopic varices at different anatomical sites. For example, a retrospective cross-sectional study conducted in Mexico identified rectal varices as the most common form of ectopic varices, a finding that aligns with the results of several Japanese studies [16,17,18]. In contrast, studies by Lebrec and Benhamou have reported SB varices as the most common ectopic varices, a finding consistent with the results of our study [19]. These differences in the localization of ectopic varices may be influenced by variations in the underlying causative disorders, patient populations, or diagnostic focus. Notably, our study specifically targeted patients undergoing endoscopic evaluation for suspected SB bleeding. Consequently, the predominance of SB varices in our study reflects the unique characteristics and clinical context of this specific patient group.

Ectopic varices account for up to 5% of all variceal bleeding cases, with SB varices being a particularly rare subset [1]. In our study, SB varices were diagnosed at an extremely low frequency (0.01%), reflecting their rarity and the challenges associated with their detection. A review of 169 cases of ectopic variceal bleeding reported that 17% involved the duodenum and another 17% involved the jejunum and ileum, highlighting the small bowel as a significant, albeit uncommon, site of ectopic varices [5]. The Japan Society for Portal Hypertension reported in their findings that 39.3% of ectopic varices were located in the SB, further subdivided into 32.9% in the duodenum, 4% in the jejunum, 1.2% in the ileum, and 1.2% in an unidentified part of the small intestine [17]. Additionally, a study conducted in Italy found SB varices in 8.1% of patients undergoing capsule endoscopy; these patients had cirrhosis and portal hypertension [20]. Similarly, a Japanese study identified SB varices in 7% of patients with portal hypertension via capsule endoscopy, and another study reported an 8.7% prevalence among patients undergoing the same diagnostic modality [21,22]. In contrast, our study, which relied exclusively on DBE for diagnosis, identified fewer cases of SB varices than studies using radiology or capsule endoscopy. The differences in prevalence rates highlight the impact of the diagnostic modality on the detection of SB varices. The serosal and submucosal locations of varices, coupled with the limitations of DBE in visualizing these layers, likely contributed to the lower detection rate in our study. Furthermore, as a high-volume reference center for small bowel disease, the large number of DBE cases analyzed may have diluted the observed frequency of SB varices, emphasizing the need for complementary diagnostic approaches to achieve a more comprehensive evaluation.

While guidelines for the management of gastric and esophageal varices are well established [23], there are no standardized treatment protocols for SB varices owing to their rarity and limited data available. The treatment options for SB variceal hemorrhage encompass a range of approaches, including conservative management, radiological interventions, pharmacological therapy, endoscopic procedures, and surgical treatments. The Baveno VII consensus suggests that, in patients with ectopic varices, either endovascular or endoscopic treatment options should be considered [24]. DBE has proven to be a valuable tool for diagnosing and treating SB varices [25]. Endoscopic treatment, particularly in experienced centers, is a minimally invasive option that is influenced by the location of the ectopic varices, expertise of the endoscopist, and available therapeutic techniques. A Japanese study involving 11 patients with SB varices reported endoscopic injection therapy in one case, while none underwent band ligation alone or in combination with injection therapy [17]. Duodenal varices in a study were managed with injection therapy or a combination of band ligation and injection [13]. In our study, endoscopic treatment strategies included endoscopic injection sclerotherapy in eight cases, cyanoacrylate used in six cases, and polidocanol in two. Additionally, one patient was treated with a combination of endoscopic injection sclerotherapy and hemoclipping. Notably, no cases of band ligation were reported in our cohort. Various sclerosant agents, such as cyanoacrylate and polidocanol, have been utilized for endoscopic injection therapy, and DBE-guided cyanoacrylate injection therapy has been documented in several case reports and studies as a successful approach for treating SB varices [13,14,25,26]. Successful treatment of SB varices using endoscopic banding has also been reported, albeit less frequently [11]. The first choice for hemostatic therapy was cyanoacrylate, given its rapid polymerization and effectiveness in controlling active bleeding. However, in cases where cyanoacrylate was not readily available during double balloon enteroscopy, polidocanol was used as an alternative. This sclerosing agent is a viable option for managing bleeding varices in an emergency setting. The decision to use polidocanol was based on its availability and the need for prompt hemostasis. These findings underscore the importance of tailoring the treatment approach to the individual patient, considering the anatomical and clinical characteristics of the varices and the expertise of the treatment center. Further studies are needed to establish standardized guidelines for the management of SB varices and evaluate the long-term outcomes of various therapeutic modalities.

This study had several limitations that should be acknowledged. First, the sample size was relatively small, with only 11 patients identified as SB varices over an 18-year period. This limited sample size reduced the generalizability of the findings and highlighted the rarity of this condition. Second, the retrospective design of the study inherently restricts the ability to establish causality and makes it prone to selection and information bias. Additionally, the study was conducted at a single tertiary referral center, which may have introduced referral bias and limited the applicability of the results to other settings or broader patient populations. Another limitation is the exclusive reliance on DBE for the diagnosis and treatment of SB varices. While DBE is a valuable diagnostic and therapeutic tool, it may not effectively detect serosal or submucosal varices, potentially underestimating the prevalence of SB varices. Furthermore, the absence of a control group prevents direct comparison of the outcomes of DBE with other diagnostic or therapeutic modalities, such as capsule endoscopy, radiological imaging, or surgical interventions.

## 5. Conclusions

Small bowel varices represent a potentially life-threatening cause of gastrointestinal bleeding, with diagnostic challenges often complicating timely hemostasis. Double balloon enteroscopy has emerged as a pivotal tool in diagnosing and managing small bowel varices, offering endoscopists the ability to visualize and intervene in these rare cases directly. Factors such as the size and location of the varices, as well as the expertise of the treatment center, play critical roles in determining therapeutic success. In our findings, the proximal small bowel was identified as the most common site for variceal bleeding, underscoring the importance of carefully selecting the appropriate double balloon enteroscopy route—oral or anal—to optimize the diagnostic and therapeutic outcomes. This route selection is particularly crucial given the anatomical complexity of the small bowel and the variability in variceal location. With its ability to provide targeted access and therapeutic intervention, double balloon enteroscopy remains an invaluable tool in addressing the challenges posed by small bowel varices. Less invasive diagnostic methods such as capsule endoscopy can be used in the follow-up of these patients. Further research is needed to refine diagnostic strategies and develop standardized treatment protocols for these rare but serious lesions.

## Figures and Tables

**Figure 1 diagnostics-15-00336-f001:**
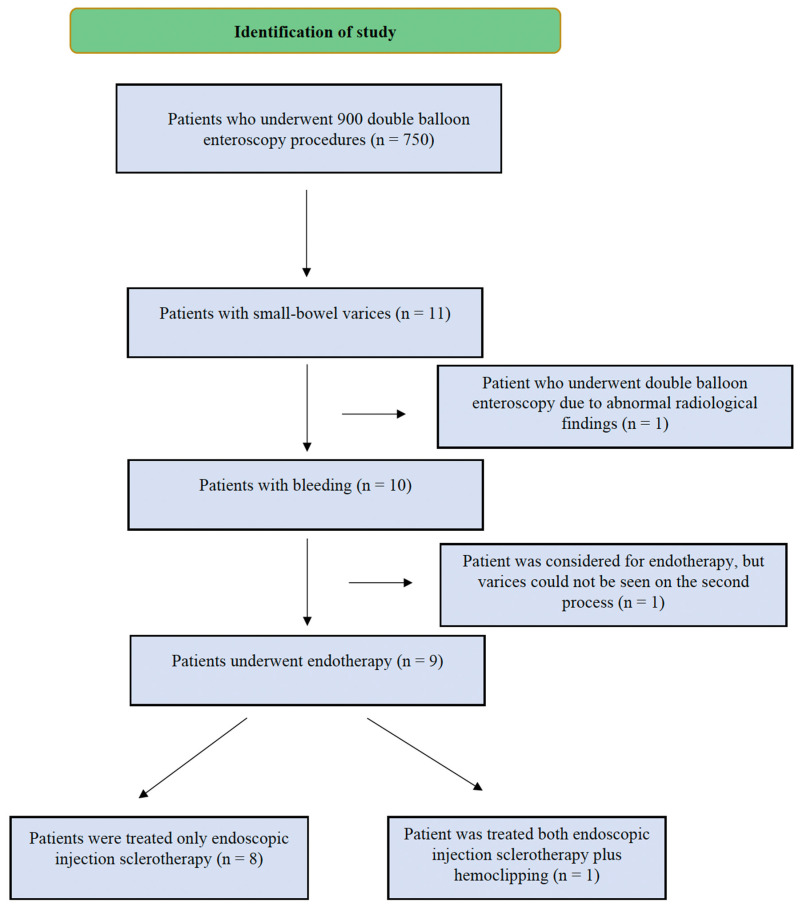
The flowchart of the search strategy.

**Figure 2 diagnostics-15-00336-f002:**
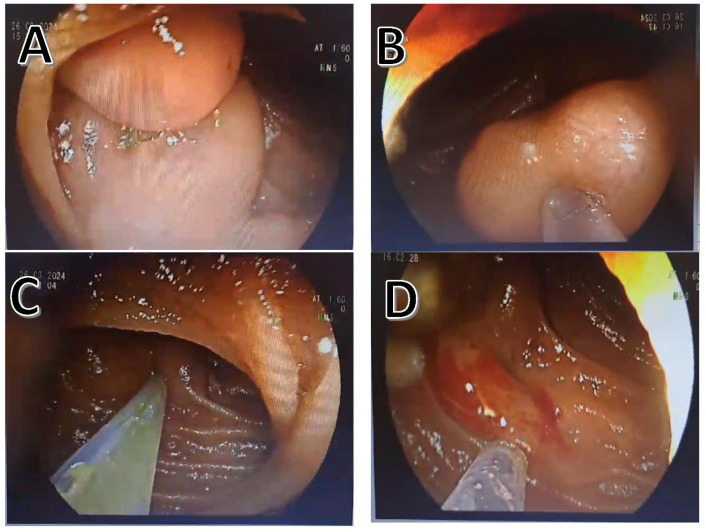
Endoscopic images depicting treatments applied to small bowel varices. (**A**) Small bowel varix; (**B**–**D**) small bowel varix injection sclerotherapy.

**Table 1 diagnostics-15-00336-t001:** Etiology of small bowel varices diagnosed by double balloon enteroscopy.

Etiology	No. of Patients, *n*(%)
Postsurgical adhesion	
Gastrojejunostomy	1(9.1%)
Hepaticojejunostomy	1(9.1%)
Subtotal colectomy	1(9.1%)
Portomesenteric venous thrombosis	
Liver transplantation	2(18.2%)
Pancreas cancer	1(9.1%)
Pancreatitis	1(9.1%)
Thrombophilia	1(9.1%)
Unknown	1(9.1%)
Cirrhosis	1(9.1%)
Unknown	1(9.1%)

**Table 2 diagnostics-15-00336-t002:** Small bowel varices associated with portomesenteric venous thrombosis and other conditions.

Patient	Etiology	Condition	Additional Details
Case-1	Surgery	Gastrojejunostomy	Gastric outlet obstruction
Case-2	Surgery	Hepaticojejunostomy	Liver alveolar hydatid cyst
Case-3	Unknown	-	-
Case-4	Portomesenteric Venous Thrombosis	Liver Transplantation	Hepatitis B Virus, only portal vein thrombosis
Case-5	Surgery	Subtotal Colectomy	Diverticulosis
Case-6	Portomesenteric Venous Thrombosis	Unknown	Only mesenteric vein thrombosis
Case-7	Portomesenteric Venous Thrombosis	Pancreatic Cancer	Portal and mesenteric vein thrombosis
Case-8	Portomesenteric Venous Thrombosis	Pancreatitis	Only mesenteric vein thrombosis
Case-9	Portomesenteric Venous Thrombosis	Liver Transplantation	Alcohol, only portal vein thrombosis
Case-10	Portal Hypertension	Cirrhosis	Hepatitis B Virus
Case-11	Portomesenteric Venous Thrombosis	Thrombophilia	Portal and mesenteric vein thrombosis

**Table 3 diagnostics-15-00336-t003:** Features of double balloon enteroscopy treatment.

Patient	DBE Procedures	DBE Indication	DBE Diagnosis	DBE Route	Location	Treatment	Treatment Plan	Results
Case-1	1	Overt bleeding	Varix	Oral	Jejunum	Polidocanol	Emergent	Not bleeding
Case-2	1	Overt bleeding	Adherent clot	Oral	-	-	Emergent	Abundant massive bleeding, exitus
	2	Overt bleeding	Varix	Oral	Jejunum	Cyanoacrylate	-	
	3	Overt bleeding	Adherent clot	Oral	-	-	-	
	4	Overt bleeding	Adherent clot	Oral	-	-	-	
	5	Overt bleeding	Varix	Oral	Jejunum	Cyanoacrylate	-	
	6	Overt bleeding	Adherent clot	Anal	-	-	-	
	7	Overt bleeding	Varix	Oral	Jejunum	Cyanoacrylate		
	8	Overt bleeding	Varix	Oral	Jejunum			
Case-3	1	Overt bleeding	Varix	Oral	Jejunum	-	Prophylactic	Not bleeding, varix appeared
	2	Treatment	Varix	Oral	Jejunum	Cyanoacrylate	-	
	3	Occult bleeding	Varix	Oral	Jejunum	-	-	
Case-4	1	Overt bleeding	Varix	Oral	Jejunum	-	Prophylactic	Not bleeding, varix disappeared
	2	Treatment	Varix	Oral	Jejunum	Cyanoacrylate	-	
	3	Occult bleeding	Normal	Oral	-	-	-	
Case-5	1	Overt bleeding	Varix	Oral	Jejunum	Polidocanol	Emergent	Not bleeding, varix disappeared
	2	Occult Bleeding	Normal	Oral	-	-	-	
Case-6	1	Overt bleeding	Varix	Oral	Jejunum	Cyanoacrylate	Emergent	Not bleeding, at two different locations
	2	Overt bleeding	Varix	Oral	Jejunum	Cyanoacrylate	-	
Case-7	1	Overt bleeding	Varix	Oral	Duodenum	Polidocanol + Hemoclipping	Emergent	Bleeding within one month
	2	Overt bleeding	Varix	Oral	Duodenum	Polidocanol	-	
Case-8	1	abnormal radiologi-cal imaging	Varix	Anal	Ileum	-	-	No treatment
Case-9	1	Overt bleeding	Varix	Oral	Ileum	Cyanoacrylate	Emergent	Not bleeding, varix disappeared
	2	Occult bleeding	Normal	Oral	-	-	-	
Case-10	1	Overt bleeding	Varix	Oral	Ileum	-	-	Prophylactic endotherapy was considered, not detected on second look
	2	Treatment	Normal	Oral	-	-	-	
Case-11	1	Overt bleeding	Varix	Oral	Jejunum	Cyanoacrylate	Emergent	Bleeding within one month
	2	Overt bleeding	Varix	Oral	Jejunum	Cyanoacrylate	-	

## Data Availability

The datasets used and/or analyzed in this study are available upon reasonable request from the corresponding author.

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
