# Peer review of "Effectiveness of Double Balloon Enteroscopy in the Diagnosis and Treatment of Small Bowel Varices"

_diagnostics, 2025, doi:10.3390/diagnostics15030336_

Round 1

Reviewer 1 Report

Comments and Suggestions for Authors

Dolu and colleagues provided an interesting manuscript on the enteroscopic management of SB varices, a potentially life-threatening entity. Considering the rarity of this condition, the described population appears to be large compared to similar studies in recent literature. Moreover, the observation timeframe provides interesting insights on the long-term follow-up of these patients.

While praising the Authors' work, there are some point that need to be addressed:

- in the introduction, please consider using the term "device-assisted enteroscopy (DAE)" before introducing DBE, for clarity

- it is not very clear why 4 patients repeated control DBE procedures, considering that it is stated that in 10 patients with 2-years follow-up no recurrence of GI bleeding occurred. In case these procedures were performed as usual follow-up (no bleeding signs), which was the timing for performing repeat DBE?As a suggestion to the Readers, could SBCE be a viable, less invasive alternative? -> I suggest editing Table 3, perhaps adding a column "DBE indication" before "DBE diagnosis"...this would highly enhance readability.

- In the Discussion, a brief comment on how to choose the hemostatic modality (for instance, polidocanol vs cyanoacrylate) would be extremely valuable for the Readers

Author Response

First of all, we would like to very thank reviewer#1 for taking her/his valuable time to evaluate our manuscript and for her/his valuable comments. Your comments have greatly contributed and improved our paper, and its quality has increased significantly. 

1- in the introduction, please consider using the term "device-assisted enteroscopy (DAE)" before introducing DBE, for clarity

It has been corrected and highlighting in yellow.

  1. it is not very clear why 4 patients repeated control DBE procedures, considering that it is stated that in 10 patients with 2-years follow-up no recurrence of GI bleeding occurred. In case these procedures were performed as usual follow-up (no bleeding signs), which was the timing for performing repeat DBE?As a suggestion to the Readers, could SBCE be a viable, less invasive alternative? -> I suggest editing Table 3, perhaps adding a column "DBE indication" before "DBE diagnosis"...this would highly enhance readability.

****The ten surviving patients had a median follow-up of two years, during which no recurrent gastrointestinal bleeding was observed (repeat control DBE procedures were conducted in four patients, with three ectopic varices disappearing post-treatment).

***Patients who had stigmata but no active bleeding underwent endoscopic treatment in another session because histoacryl was not available in our hospital at the time of diagnosis. Therefore, some patients underwent repeated enteroscopy after histoacryl was obtained. However, the patients with active bleeding underwent endoscopic treatment with polidocanol. In addition, when DBE was performed in these patients, capsule endoscopy could not be used for diagnostic purposes because it was not available in our hospital. Patients with suspected small-bowel bleeding underwent DBE again.

Related infotmation has been added in the manuscript and highlighting in yellow

***Less invasive diagnostic methods such as capsule endoscopy can be used in the follow-up of these patients.

Related infotmation has been added in the manuscript and highlighting in yellow

- In the Discussion, a brief comment on how to choose the hemostatic modality (for instance, polidocanol vs cyanoacrylate) would be extremely valuable for the Readers

***The first choice for hemostatic therapy was cyanoacrylate given its rapid polymerization and effectiveness in controlling active bleeding. However, in cases where cyanoacrylate was not readily available during double-balloon enteroscopy , polidocanol was used as an alternative. This sclerosing agent is a viable option for managing bleeding varices in an emergency setting. The decision to use polidocanol was based on its availability and the need for prompt hemostasis.

Related infotmation has been added in the manuscript and highlighting in yellow

Reviewer 2 Report

Comments and Suggestions for Authors

dear authors

the issue illustated in your paper is really interesting and challenging.

some m inor revisions:

<introduction is too long

materials and method:

just explain more clearly the origin of varices (post surgical adhesions???)

and explain more clearly how do you rach diagnosis (radiological by entero-ct scan??') ; any role for capsule endoscopy?

In page 3 the paragraph from line 5 to 15 in not useful and confusing.

explain better the technique and the amount of sclerosing agent used for each procedure

In the results are reported two cases of death due to the bleeding<, explain if they were submitted to alternative therapies (radiological, surgical) or why not

The discussion and references should be implemented by adding Baveno guidelines

Author Response

First of all, we would like to very thank reviewer#2 for taking her/his valuable time to evaluate our manuscript and for her/his valuable comments. Your comments have greatly contributed and improved our paper, and its quality has increased significantly. 

introduction is too long

***I agree that the introduction segment is somewhat long; however, it needed at least 3,500 words to comply with the journal's requirements, and I felt that all the information included was necessary for the article's comprehensibility.

materials and method:

just explain more clearly the origin of varices (post surgical adhesions???)

***Patients were further stratified based on the underlying etiology of SB varices and classified into four categories: postsurgical adhesion, portomes-enteric venous thrombosis, cirrhosis, and unknown etiology. Importantly, patients with portomesenteric venous thrombosis were excluded from any other etiological group to avoid overlap. All patients underwent thorough radiological and laboratory evaluations to assess portal hypertension, with and without accompanying portomesenteric venous thrombosis. Given that prior abdominopelvic surgeries can predispose individuals to the formation of atypical collaterals, leading to ectopic varices [1], the potential role of previous abdominopelvic surgeries was carefully evaluated as an additional etiological factor. This multifaceted analysis aimed to elucidate the diagnostic and therapeutic chal-lenges associated with SB varices and contribute meaningful data to the limited body of literature on this subject.  

Related infotmation has been added in the manuscript and highlighting in yellow

****Intra-abdominal adhesions from previous abdominal surgeries can predispose patients to the development of collaterals in unusual locations, leading to ectopic varices.

and explain more clearly how do you rach diagnosis (radiological by entero-ct scan??') ; any role for capsule endoscopy?

In this study, double-balloon enteroscopy  played a central role as a diagnostic and therapeutic tool. However, radiological modality, such as CT or CT angiography, was often utilized as complementary tools to identify varices and their anatomical locations.  In addition capsule endoscopy could not be used for diagnostic tools because it was not available in our hospital. The study population comprised patients who underwent DBE for a range of indications, including gastrointestinal bleeding and abnormal findings on radiological imaging. In this study, we specifically included patients with SB varices diagnosed using DBE.

Related infotmation has been added in the manuscript and highlighting in yellow

In page 3 the paragraph from line 5 to 15 in not useful and confusing.

****Our hospital serves as a tertiary referral center for DBE, attracting a diverse patient population with various gastrointestinal conditions. The study population comprised pa-tients who underwent DBE for a range of indications, including gastrointestinal bleeding and abnormal findings on radiological imaging. In this study, we specifically included patients with SB varices diagnosed using DBE. Patients who did not undergo DBE or were under 18 years of age were excluded to ensure a focused and consistent analysis of this unique cohort. This selection criterion allowed for a comprehensive evaluation of the di-agnostic and therapeutic roles of DBE in managing SB varices.

It has been corrected in the manuscript and highlighting in yellow

Explain better the technique and the amount of sclerosing agent used for each procedure

***In all cases, SB varices were diagnosed using DBE a highly specialized endoscopic technique. The DBE system used for the procedures consisted of an overtube and enter-oscope, pumping unit, and main computer. All procedures were performed using the Fu-jinon EN-450T5 enteroscope (Fujinon Corp., Saitama, Japan), a device specifically de-signed for deep and comprehensive visualization of the small bowel.

****For the endoscopic treatment of varices, sclerosing agents such as cyanoacrylate and polidocanol were administered under direct visualization. The exact amount of the agent used varied based on the size, location, and number of varices. Cyanoacrylate was typically injected in volumes ranging from 0.5 to 1.0 mL per varix, diluted with lipiodol to ensure adequate radiopacity and effective distribution within the varix. Polidocanol, on the other hand, was administered as a 1% solution in volumes of 1–2 mL per varix, depending on its dimensions and extent of bleeding. After the injection, the treated area was inspected for hemostasis, and additional measures, such as hemoclipping, were applied if needed.

Related infotmation has been added in the manuscript and highlighting in yellow

In the results are reported two cases of death due to the bleeding<, explain if they were submitted to alternative therapies (radiological, surgical) or why not

*** In our study, only one patient died of severe bleeding due to small-bowel varices despite undergoing  endoscopic intervention. This patient had no alternative therapies such as radiological or surgical interventions. This case involved significant hemodynamic instability, which precluded a safe transfer to interventional radiology. Furthermore, the patient’s  poor overall prognosis led  to a focus on endoscopic treatment rather than high-risk surgery.

Related infotmation has been added in the manuscript and highlighting in yellow

The discussion and references should be implemented by adding Baveno guidelines

*** The Baveno VII consensus suggests that, in patients with ectopic varices, either endovascular or endoscopic treatment options should be considered.

Related infotmation has been added in the manuscript and highlighting in yellow